# Ankle Somatosensation and Lower-Limb Neuromuscular Function on a Lunar Gravity Analogue

**DOI:** 10.3390/brainsci15050443

**Published:** 2025-04-24

**Authors:** Ashleigh Marchant, Nick Ball, Jeremy Witchalls, Sarah B. Wallwork, Gordon Waddington

**Affiliations:** 1Research Institute for Sport and Exercise, University of Canberra, Canberra, ACT 2617, Australia; 2IIMPACT in Health, University of South Australia, Adelaide, SA 5001, Australia

**Keywords:** hypogravity, active movement extent discrimination assessment (AMEDA), lower-limb muscle activity, electromyography (EMG), Myoton, muscle tone, muscle stiffness, wedge bed

## Abstract

**Background/Objectives**: The adverse effects of low gravity on human physiology are well documented; however, much of the literature is directed at changes which occur in microgravity (µg: weightlessness) with relatively less documented on changes in hypogravity (<1 g; >µg: gravity less than Earth’s but more than microgravity). Somatosensation and neuromuscular control may be of particular importance for astronauts as they prepare for future missions to walk on the Moon. This study aimed to explore the effect of reduced weight bearing (to simulate conditions of hypogravity) on ankle somatosensation, lower-limb muscle activity, tone, and stiffness, compared to full weight bearing. **Methods**: Participants completed an ankle somatosensory acuity task (active movement extent discrimination assessment [AMEDA]) in two body positions: (1) upright standing (1 g), and (2) in a head-elevated supine, semi-weight bearing (0.16 g) position using a custom-built inclined “wedge bed”. The second position induced ~16% body weight on to the plantar aspect of the feet, simulating that of lunar gravity. We compared the AMEDA scores between the two positions. Lower-limb muscle activity was recorded via surface EMG throughout the AMEDA task for both positions. The ankle AMEDA has five depths of ankle inversion. We compared muscle activity between the body positions, and muscle activity between inversion depths “1” and “5” (within each position). Lower-limb muscle tone and muscle stiffness were assessed at rest in both body positions using the MyotonPRO. **Results**: Fifty-five participants between the ages of 18 and 65 (28 females, 27 males; mean age of 40 years) completed the study. The AMEDA scores, muscle tone and stiffness were reduced when the participants were on the lunar wedge bed, compared to upright standing (*p* = 0.002; *p* < 0.001; *p* < 0.001). Some lower-limb muscles exhibited less activity in the lunar wedge-bed position compared to upright standing (tibialis anterior, peroneus longus, vastus lateralis, rectus femoris; *p* < 0.05) but others were unchanged (gastrocnemius, vastus medialis; *p* > 0.05). Muscle activity was unchanged between the AMEDA depths (*p* = 0.188). **Conclusions**: The results provide insight into how the somatosensory and neuromuscular systems respond to reduced weight bearing and potentially lunar gravity conditions, thereby informing how to target interventions for future missions.

## 1. Introduction

It is well documented that exposure to low gravity can have unfavorable effects on human physiology, including reduced bone density, muscle atrophy, decreased cardiovascular fitness, cerebral cortex reorganization, and reduced cognitive function [1,2,3]. Completing various tasks in such unique conditions significantly challenges human performance and errors can have catastrophic outcomes, such as spacesuit damage resulting from a fall, or an astronaut’s inability to effectively control a vehicle while exploring the lunar surface [4,5,6]. These changes in physiological systems experienced by astronauts are significant and often contribute to poor health, so gaining deeper insights into these changes, through simulation techniques on Earth, is important for the safety and success of astronaut missions, particularly as space agencies prepare to explore the Moon in the near future [4,7].

While physiological changes in microgravity have been relatively well explored, this study will explore the changes in hypogravity, which remain comparatively less understood [4,8,9,10,11]. Microgravity (µg) is near zero gravity characterized by weightlessness. Hypogravity (<1 g; >µg) is gravity less than Earth’s but more than microgravity, such as that on the Moon [4,12]. This knowledge gap is particularly significant considering the relevance of hypogravity with the current interest in sending humans to the Moon and Mars [11,13,14]. Notably, the lack of studies largely stems from the limited number of astronauts who have been exposed to these conditions. To date, only twelve Apollo astronauts have walked on the Moon [15,16]. The high frequency of falls experienced by crew members after missions to the Moon suggests a significant reduction in physical capability, with mobility issues often persisting after prolonged exposure to low gravity environments [9,17]. This may be attributed to physiological changes occurring which limit function, or it could be an indication of crew members navigating their new environment [18]. The gravity level at which these changes might first manifest remains unclear.

Limitations in space experimentation arise due to the novelty of the research area, small sample sizes, and high costs [19]. Consequently, many studies aim to replicate the effects of micro- and hypogravity whilst accommodating for the constraints of 1 g (Earth’s gravity). Since the gravitational acceleration cannot be altered on Earth, researchers often aim to increase or decrease a person’s mass to alter the relevant weight, thereby imitating a change in gravity [17,20]. Common methods to alter one’s weight include parabolic flights, dry immersion, human centrifuges, lower body positive pressure (LBPP), and bed rest [17,21]. Some of these methods are beneficial in addressing acute disrupted function (e.g., immediate physiological changes and astronaut adaptation to a novel environment). Others capture the long-term effects of reduced gravity, often concerning the impact of disuse. Bed rest in particular offers significant experimental utility and can be administered for short or long durations to induce physiological changes like those experienced in space travel, as prolonged physical inactivity leads to comparable human physiologic degradation [22]. Extensive bed rest involves maintaining a supine position for several days or weeks and is thought to simulate physiological deterioration similar to low gravity as the body struggles to biologically re-orientate itself [23]. However, the high resource costs associated with such experimental designs can limit feasibility, especially when results are based on small sample sizes [24]. Alternatively, a simpler approach to address acute changes in performance involves shifting the body from upright to supine and assessing immediate changes [25]. These techniques can be beneficial for capturing a broad range of participants efficiently at little resource cost and identifying acute changes in performance. Even a rapid change in gravity is sufficient to alter motor commands, even if only partially [4,26].

Proprioception, also referred to as joint position sense, contributes with tactile sensation to form somatosensation [27]. Somatosensation plays a vital role in maintaining an upright, stable posture by interpreting gravity to influence and assist in motor planning [26,28]. Gravity is therefore essential for numerous sensory cues and establishes a sense of verticality. However, the impact of altered gravity environments on somatosensation is less understood. Astronauts mobilizing on the Moon exhibit a loss of balance, lack of spatial awareness, and excessive body tilt [17]. They will often adopt a jumping- or skipping-style of locomotion as opposed to the typical stance–swing gait cycle [29]. Our previous acute bed-rest study, which simulated acute microgravity conditions (~0 g) at the foot and ankle, revealed a significant reduction in ankle somatosensation during supine lying [25]. Other research suggests a general disorientation occurs between 0.16 g and 0.38 g (the Moon’s and Mars’ gravity, respectively), with object manipulation not being affected until gravity drops below 0.2 g [4]. It is plausible that changes to somatosensation have played a critical role in astronauts’ unusual gait pattern when walking on the Moon, yet studies often assess sensorimotor function without specifically addressing somatosensation [1,30]. This poses challenges in discerning the extent of proprioceptive changes that may have occurred and at what “g” exposure. Greater knowledge in this domain can guide interventions aimed at mitigating the adverse effects of altered gravity on sensory function.

Muscle function is intricately intertwined with somatosensation, forming an integral part of the sensorimotor system [31,32]. In true and simulated low gravity, patterns of muscle contraction are affected, and biomechanical properties are compromised. For example, in simulated microgravity, peak muscle activity is lowered, contraction time is increased for major lower leg muscles [25], and muscle tone is reduced in both trunk and lower-limb musculature [33]. In a real microgravity setting, muscle stiffness is shown to decline over time in the tibialis anterior and soleus despite countermeasures being set in place [22]. This might be due to changes in muscle activity at immediate exposure [25] or reductions in muscle size and strength over time [1], but the direct correlation with somatosensory function, particularly in hypogravity, is less clear. Our previous acute bed-rest study not only demonstrated that ankle somatosensation is reduced in the flat supine position when compared to upright standing, it also showed a reduction in peak muscle activity for the tibialis anterior, peroneus longus, and gastrocnemius, during the somatosensory acuity task in the supine condition [25]. Assessing muscle activity and soft tissue dynamics during such tasks is therefore crucial for determining optimal targets for exercise regimes [22]. Knowledge of the required level of “g” for the maintenance of muscle health or the extent of muscle deterioration which might occur at 0.16 g is imperative, particularly as future missions plan to establish habitation on the Moon [17]. Previous research in simulated hypogravity has shown differences in gait patterns and gastrocnemius muscle contractile behavior between simulated Moon (0.16 g) and Mars (0.38 g) gravity [34]. Additionally, while motor patterns remain consistent during bouncing motions across various “g” levels, lower-limb muscle activation adapts depending on the gravity, and muscle fascicle behavior is regulated differently in drop-landings, also dependent on the gravity [27,35]. This suggests that, while muscle changes are expected in response to low gravity, the alterations vary across different gravitation levels and are not equivalent. Further understanding of how neuromuscular function responds to these initial changes in gravity, particularly during proprioceptive tasks, could provide valuable insight into muscle health, informing the development of more effective pre- and post-rehabilitation strategies. In the short term, rapid changes in gravity can be utilized to assess characteristics such as muscle tone and stiffness, which may serve as early indicators of potential disorders [22].

The primary aim of the present study was to explore the effect of reduced weight bearing (head-elevated supine lying) on ankle somatosensation compared to full weight bearing (upright standing). Ankle somatosensation was measured via active movement extent discrimination assessment (AMEDA) in two body positions: (1) upright standing with full weight-bearing (1 g) position and (2) head-elevated supine, semi-weight-bearing position (mimicking 0.16 g conditions) using a custom-built inclined “lunar wedge bed”. We hypothesized that ankle somatosensory acuity (as measured by the AMEDA) would be reduced in the head-elevated supine-lying lunar wedge-bed condition compared to upright standing. The secondary aim of this study was to explore the effect of reduced weight bearing on lower-limb neuromuscular function compared to full weight bearing. Peak lower-limb muscle activity (measured via surface EMG during the AMEDA task) and lower-limb muscle tone and muscle stiffness (both measured via the MyotonPRO at rest) were assessed in both positions. We hypothesized that each of these assessments would be impaired in the lunar wedge-bed 0.16 g condition, when compared to upright standing. Another secondary aim was to explore how muscle activity differed during the high and low depths of ankle inversion when completing the AMEDA (i.e., stop 1 vs. stop 5) in both positions. We hypothesized that an increase in AMEDA platform stop (i.e., greater depths of ankle inversion) would be associated with an increase in peak muscle activity in both positions.

## 2. Materials and Methods

### 2.1. Participants

Fifty-five healthy participants between the ages of 18 and 65 were recruited for this study. Based on our previous study and a medium effect size of 0.69, data analytic software, G*Power (version 3.1.9.7; RRID:SCR_013726), was used to determine that we required at least 54 participants to achieve a statistical power of 0.80 [25]. The study was approved by the University of Canberra Human Research Ethics Committee (reference number: 202312043). The study protocol was uploaded to the Open Science Framework prior to data collection [36]. Preparation of this manuscript was completed using Microsoft Word (Microsoft^®^ Corporation, Redmond, WA, USA. 2021. Microsoft® Word. version 16. Retrieved from https://www.microsoft.com/word (accessed on 30 October 2021)). Recruitment was via media advertisement (paper flyers, email, radio) and word of mouth. Inclusion criteria were adults between 18 and 65 years, who could understand and speak English, and considered themselves healthy and unrestricted (i.e., had the ability to move without any restrictions that impacted day-to-day tasks). This was to ensure they had the ability to complete the ankle AMEDA task and move between positions without difficulty. Exclusion criteria were any conditions which may have affected balance (e.g., diminished sensation, inner ear loss of function) or an ankle injury within the previous three months.

### 2.2. Lunar Wedge Bed

To mimic the conditions of lunar gravity (0.16 g), the lunar wedge bed was custom built for this study to apply an axial load of 16% of the participants’ body weight. Based on our previous study where somatosensory acuity was assessed in a lying supine position (horizontal) [25], the lunar wedge bed was designed to investigate somatosensation under “head-elevated” supine or partial lower-limb loading. The timber truss-style system involved a timber board positioned at 9.2 degrees up from horizontal. In this set-up, gravity and mass were not directly reduced. Instead, forces were applied elsewhere in the body to reduce the axial load or force exerted under the feet in contact with the ground [20,21]. An angle of 9.2 degrees was chosen to simulate the conditions of lunar gravity (0.16 g) under the feet, based on the following calculation:FN = FWSin θ
where FN represents the normal force (1 g) acting perpendicular to the bed surface and FW was the gravitational force pulling the participant downwards (0.16 g) [37]. The angle of 9.2 degrees was selected to achieve an axial load corresponding to 16% of the participant’s body weight. For example, if a participant weighs 100 kg (1 g), they will experience an axial load of 16 kg (0.16 g) through the feet when their head is elevated at 9.2 degrees above the horizontal. This angle was determined by considering the vertical component of the applied force necessary to replicate the reduced gravitational force of the Moon, ensuring the correct proportion of body weight is redistributed along the lunar wedge bed’s surface. The lunar wedge bed was selected for this study over other common methods used to simulate hypogravity due to its ease of use and ability to directly assess the lower-limb somatosensory system independent to the visual and vestibular systems. Proprioceptive tasks in supine studies have been shown to positively correlate with upright functional tasks [38]. As a result, this set up is an ideal method for somatosensory research, as they allow assessment of the somatosensory system without affecting the vestibular system [38]. This enables direct assessment of the response of the somatosensory system to body unloading, independent of vestibular changes associated with spaceflight, and with minimal discomfort for the participant [21].

Participants were lying supine on a traditional mechanics’ creeper, which had six wheels and was placed on top of the timber board, which allowed them to slide and position their feet to settle onto the AMEDA platform with minimal friction. The AMEDA platform was set at the base of the timber board at 90° to the bed angle. In this position, participants had sufficient pressure on the plantar aspect at their feet at ~16% of their body weight, to simulate the pressure under their feet of 0.16 g (Moon gravity). Participants were in this position for an acute period only, which was approximately 10 min or the duration of having the MyotonPro biomechanical properties assessed and full completion of the AMEDA. See Figure 1 for the participant lunar wedge-bed setup.

### 2.3. Outcome Measures

The outcome measures used for this study were ankle somatosensory acuity and lower-limb neuromuscular function in both positions (upright standing and head-elevated supine lying on the lunar-wedge bed). Ankle somatosensory acuity was measured via the ankle AMEDA. Lower-limb neuromuscular function was divided into two: muscle activity (via EMG) and muscle biomechanical properties (muscle tone and muscle stiffness via the MyotonPRO; Myoton AS, Tallinn, Estonia).

#### 2.3.1. Active Movement Extent Discrimination Assessment (AMEDA)

The AMEDA task was used to assess participants’ ankle somatosensory acuity. The task required participants to place both feet on the AMEDA, either in (1) upright standing position or (2) while lying on the lunar wedge bed. Participants were asked to look straight ahead (perpendicular to their body angle) to reduce visual cues about their ankle position and to ensure that they were using proprioceptive and tactile feedback to complete the task. Participants completed the assessment on their non-dominant foot, determined by their non-kicking leg, as this has been demonstrated to have greater sensitivity to proprioceptive tasks [39]. The non-testing foot was placed on the side with the stationary platform while the testing foot was placed on the side with the moveable platform (facilitating various degrees of ankle inversion). During the task, participants were asked to actively move the platform (testing foot) by inverting their ankle as far as the platform would allow (solid endpoint) and then return the platform to the neutral position. The task had five pre-defined depths of ankle inversion, with each being approximately 1 degree of difference between each stop (max 4 degrees between depth 1 and 5; depth 1: 10.5° from horizontal, depth 5: 14.5° from horizontal). These depths have been used previously in ankle somatosensation research and were deemed an appropriate challenge for quantifying an individual’s somatosensory acuity while performing the inversion movement at the mid-range of ankle inversion, suitable for assessing somatosensation during active body movements [40,41,42]. Before completing the formal assessment, participants were familiarized with the five depths of inversion. When completing the ankle inversion task, participants were asked to make an absolute judgement about their ankle joint position by reporting the “depth” of ankle excursion (1 to 5). This was completed 50 times with the inversion depths presented in a pseudorandom order (10 of each depth). The entire assessment took approximately 5 min to complete. The same AMEDA device was used for both positions (upright standing and head-elevated supine lying on the lunar wedge bed).

#### 2.3.2. Electromyography (EMG)

Lower-limb muscle activity was recorded via surface EMG using seven parallel bar silver electrodes (Delsys 8 channel Trigno wireless EMG system). Muscles assessed included tibialis anterior, peroneus longus, lateral and medial head of gastrocnemius, vastus medialis, rectus femoris, and vastus lateralis. Posterior upper leg, such as hamstring and gluteal muscles, were not recorded due to interference when lying supine on the lunar wedge bed. Electrode placement was guided by Rainoldi et al., 2004 [43], and the SENIAM guidelines (http://www.seniam.org/ (accessed on 30 August 2023)). Briefly, the anatomical guidelines are shown in Table 1. Muscle activity was recorded throughout each AMEDA test, and the electrodes remained on the participants’ legs between both positions (upright standing and head-elevated supine lying on the lunar wedge bed).

#### 2.3.3. MyotonPro

Lower-limb biomechanical properties were recorded via myotonometry using a digital palpation device, MyotonPRO (Myoton AS, Tallinn, Estonia). The MyotonPRO is a handheld, non-invasive tool which applies mechanical impulse via a small probe placed perpendicular to the muscle belly. It has demonstrated good to excellent reliability (ICC = 0.41−0.98) in assessing lower-limb muscle tone and muscle stiffness [44,45]. The tissue’s response to the impulse was recorded via the device’s accelerometer and used to produce a raw value for muscle tone and muscle stiffness. Triplescan mode was used throughout, producing a mean value from three oscillations, and was assessed in both positions at rest. Muscles assessed included tibialis anterior, peroneus longus, lateral and medial head of gastrocnemius, vastus medialis, rectus femoris, and vastus lateralis (i.e., the same seven muscles recorded for EMG activity). Measurement locations coincided with EMG electrode placement. As muscle tone and muscle stiffness were recorded at rest and not during the AMEDA, it was completed before and after EMG electrode placement. The upright standing assessment was completed just prior to EMG electrode placement, and head-elevated supine was assessed on the lunar wedge bed, after both AMEDAs were complete, and the EMG electrodes were removed.

### 2.4. Experimental Procedure

Participants attended the laboratory for a single 60 min session. Written informed consent was obtained prior to testing. Participants completed a demographic form to record their sex, age, height, weight, and preferred kicking foot. The physical assessments then commenced, including the assessments of ankle somatosensory acuity and lower-limb neuromuscular function (muscle activity, tone, and stiffness) in upright standing and in head-elevated supine on the lunar wedge bed. The order of testing was pseudorandomized, using an online random number generator (https://www.random.org/ (accessed on 30 August 2023)) ensuring an equal number of participants began with upright standing or began with supine lying on the lunar wedge bed. Participants’ lower-limb muscles were initially marked with a marker and MyotonPRO measurements were recorded in both body orientations: upright standing, and supine lying on the lunar wedge bed. EMG electrodes were then placed on the corresponding marker points, and participants completed their initial ankle somatosensory acuity test using the ankle AMEDA. Participants were offered a rest break before completing their second ankle somatosensory acuity task in the alternative body position. EMG recording was continuous throughout each AMEDA task. See Figure 1 for both position setups.

### 2.5. Data Processing

Participant responses for the ankle AMEDA were recorded manually via an Android tablet and uploaded to a Microsoft Excel spreadsheet (Microsoft^®^ Corporation, Redmond, WA, USA. 2023. Microsoft^®^ Excel. version 2301. Retrieved from https://office.microsoft.com/excel (accessed on 31 July 2023)). A matrix of responses was generated to form an area under the curve (AUC) score for each assessment. Participants ended up with two scores, one score per position (upright standing and head-elevated supine lying on the lunar wedge bed). Scores were between 0.5 (equivalent to chance) and 1.0 (a perfect score).

Raw EMG data were analyzed using EMGworks Analysis (version 4.1.3). A Butterworth notch filter was applied to remove artificial noise and a root-mean square (RMS) filter applied at a window length of 50 Hz. Peak RMS EMG was recorded for AMEDA inversion depths 1 and 5, and each posture (upright standing and head-elevated supine lying on the lunar wedge bed). AMEDA inversion depths 1 and 5 were selected to provide EMG data points representing the two extremes of the AMEDA only. An eighth electrode was included during assessment and placed on the AMEDA platform as an accelerometer. This was used to provide a timestamp of when each platform rotation commenced and ceased. Any recordings for the peak RMS EMG were taken between these two timepoints. If any EMG recording presented with unfilterable noise for any of the muscles for any inversion depth or any position, that subject was removed from the subsequent analysis.

Raw myoton data were recorded via the MyotonPRO and uploaded to a Microsoft Excel spreadsheet. Biomechanical properties measured were muscle tone and stiffness. These were characterized by the natural oscillation frequency and dynamic stiffness, measured in Hertz (Hz) and Newtons per meter (N/m), respectively. Participants ended up with twenty-eight data points from the MyotonPRO, representing two properties (tone and stiffness), seven muscles, and two positions (upright standing and head-elevated supine lying on the lunar wedge bed).

### 2.6. Statistical Analyses

SPSS statistics (IBM Corp. Released 2020. IBM SPSS Statistics for Windows, Version 27.0. IBM Corp, Armonk, NY, USA) was used to analyze all results. An alpha value of 0.05 was used to determine statistically significant results. The data were tested for outliers by inspecting boxplots, where any data points located more than 1.5 times the interquartile range (IQR) from the edge of the box (i.e., beyond the whiskers) were considered outliers, as flagged by SPSS Statistics. The data were tested for normality via a Shapiro–Wilk test. A one-way analysis of variance (ANOVA) was conducted to assess whether there was any change in AMEDA AUC upright standing scores among age groups (i.e., group “a”: 18–29 years; group “b”: 30–39 years; group “c”: 40–49 years; group “d”: 50–59 years; group “e”: 60–65 years. We also conducted an ANOVA to determine whether body mass index (BMI) influenced AMEDA scores, as high BMI can be associated with poor proprioceptive control at the ankle [46]. Participants were classified into three groups; group 1: BMI between 18.5–24.9; group 2: BMI between 25–29.9; group 3: BMI of 30 or above. A significant result would suggest age or BMI influenced somatosensory acuity at baseline (upright, standing) and would need to be considered as a covariant to interpret the overall results. Ankle AMEDA scores were tested for a learning effect via a paired *t* test by comparing results of a participant’s first and second assessment. As order of testing was randomized, a statistically significant result would suggest that prior exposure to the test had impacted scores instead of a change in position.

To address our primary aim of exploring the effect of reduced weight bearing (head-elevated supine on the lunar wedge bed) on ankle somatosensation compared to full weight bearing (upright standing), a paired *t* test was conducted to determine whether there was a statistically significant difference between the upright standing AMEDA score and the lunar wedge bed AMEDA score.

To address our secondary aim of exploring the effect of reduced weight bearing (head-elevated supine on the lunar wedge bed) on lower-limb neuromuscular function compared to full weight bearing, a three-way repeated measures analysis of variance (ANOVA) was conducted for each neuromuscular function outcome measure. The EMG data were assessed for an interaction between muscle (7 muscles), position (2 positions: upright standing and lunar wedge bed), and inversion depth of the AMEDA (2 depths: stop 1 and stop 5) (7 × 2 × 2 ANOVA). The MyotonPRO data were assessed for an interaction between muscle (7 muscles), biomechanical property (2 properties: tone and stiffness), and position (2 positions: upright standing and lunar wedge bed) (7 × 2 × 2 ANOVA).

## 3. Results

### 3.1. Study Population

Fifty-five participants were recruited and completed the study. On the premise of our primary aim of ankle AMEDA scores, there were no outliers, and the data were normally distributed (*p* = 0.78). There were no statistically significant differences in AMEDA AUC scores between the different age groups (*p* = 0.593). There were no significant differences in AMEDA AUC scores between the BMI groups (*p* = 0.725). Participants considered themselves to be unrestricted in movement, which was part of the inclusion criteria. This may explain why age and BMI, factors often considered limiting, did not influence performance in this group. There was no significant difference between the first ankle AMEDA to the second ankle AMEDA, suggesting there was no learning effect between the two AMEDA tasks (*p* = 0.09). There was a split of 28 females and 27 males, with a mean ± standard deviation (M ± SD) age, weight, height of 40 years ± 14, 748 N ± 126, and 173 cm ± 9, respectively. Most participants’ preferred kicking foot was their right (right: 49, left: 6).

### 3.2. Ankle Somatosensory Acuity: AMEDA Results

Participants exhibited significantly reduced somatosensory acuity when tested in the head-elevated supine lying lunar wedge bed position, when compared to the upright standing position, t(54) = 3.03, *p* = 0.002. Specifically, the mean ± standard deviation (M ± SD) ankle AMEDA AUC scores for upright standing and lunar wedge bed were 0.69 ± 0.05 and 0.67 ± 0.05, respectively. Figure 2 shows the mean AUC scores and confidence intervals for both positions. The reduction in AUC score suggests a slight, but significant decline in ankle somatosensory function when participants were in the lunar gravity simulated condition. A loss of somatosensory acuity may impair ability to detect changes in ground surfaces and potentially contribute to decreased balance and stability when standing or mobilizing on the Moon.

### 3.3. Muscle Activity: EMG Data

A considerable amount of variance was observed in the muscle activity data. Of the fifty-five participants, only twenty presented with a complete set of noise-free EMG muscle activity across all recorded muscles which informed the subsequent analysis. However, variability was still present among these twenty participants, as illustrated in Figure 3, with widespread confidence intervals. This variability should be considered when interpreting the following results. Mauchly’s test of sphericity indicated that the assumption of sphericity had been violated for the three-way interaction, χ^2^(2) = 85.81, *p* < 0.001. A Greenhouse–Geisser correction was therefore used to interpret the three-way repeated measures ANOVA.

There was no statistically significant main effect from the three-way interaction for muscle × position × inversion depth, F(2.71, 51.60) = 0.79, *p* = 0.49, ε = 0.45. There were no statistically significant two-way interactions for position × inversion depth, F(1, 19) = 3.18, *p* = 0.91, muscle × inversion depth, F(1.10, 37.90) = 0.86, *p* = 0.43, ε = 0.33, and muscle × position, F(2.07, 39.34) = 1.97, *p* = 0.15, ε = 0.35. The single element main effects showed a statistically significant difference for position F(1, 19) = 5.74, *p* = 0.027, but not for inversion depth F(1, 19) = 1.86, *p* = 0.19, or muscle, F(1.92, 36.48) = 2.11, *p* = 0.14, ε = 0.32. Simple comparisons with Bonferroni adjustment, showed EMG muscle activity was significantly reduced on the lunar wedge bed compared to upright standing for tibialis anterior, peroneus longus, vastus lateralis, and rectus femoris (*p* < 0.05 for all) but not for the medial and lateral head of gastrocnemius and vastus medialis (*p* > 0.05 for all). This could be due to the variability observed in the data, or the differing roles of the muscles, with the gastrocnemius and vastus medialis acting as stabilizers for the lower limb during the lunar wedge-bed testing. See Figure 3 for graphical representation of the data mean and confidence intervals. See Table 2 for the data mean ± standard deviation (M ± SD) and corresponding *p* values of the simple comparisons.

### 3.4. Muscle Biomechanical Properties: MyotonPRO Data

Muscle biomechanical properties were recorded via the MyotonPRO for the 55 participants. There were twelve outliers in the data. Two were removed due to incomplete measurements; however, the others were considered as genuine but unusual data points, given they were consistent measures for the participant. A total of 53 participants were therefore used for analyses. Some data were slightly positively skewed (between 0.5 and 1.0) but otherwise normal. Mauchly’s test of sphericity indicated that the assumption of sphericity had been violated for the three-way interaction, χ^2^(2) = 86.47, *p* < 0.001. A Greenhouse–Geisser correction was therefore used to interpret the three-way repeated measures ANOVA.

There was a statistically significant main effect from the three-way interaction between muscle × biomechanical property × position F(4.10, 213.03) = 27.06, *p* < 0.001, ε = 0.68. There was a statistically significant simple two-way interaction for position × biomechanical properties for tibialis anterior, F(1, 52) = 84.78, *p* < 0.001; peroneus longus F(1, 52) = 98.08, *p* < 0.001; medial head of gastrocnemius, F(1, 52) = 9.95, *p* = 0.003; lateral head of gastrocnemius, F(1, 52) = 31.01, *p* < 0.001, vastus medialis, F(1, 52) = 14.48, *p* < 0.001, vastus lateralis, F(1, 52) = 20.35, *p* < 0.001, and rectus femoris, F(1, 52) = 10.41, *p* = 0.002. There was a statistically significant simple main effect of position for the biomechanical property for each muscle (*p* < 0.01 for all) and so simple pairwise comparisons of positions were run for muscle tone and muscle stiffness with a Bonferroni adjustment applied. Results showed a statistically significant decrease in tone and stiffness for all muscles on the lunar wedge bed compared to upright standing. See Figure 4 for graphical representation of the data mean and confidence intervals. See Table 3 (muscle tone) and Table 4 (muscle stiffness) for the data mean ± standard deviation (M ± SD) and corresponding *p* values.

## 4. Discussion

This study aimed to explore the effect of reduced weight bearing (head-elevated supine on the lunar wedge bed) on ankle somatosensation compared to full weight bearing (upright standing) in an acute setting. We found that ankle somatosensory acuity scores, as measured via the ankle AMEDA, were significantly lower in reduced weight bearing (lunar wedge bed) when compared to upright standing. The secondary aims were to explore the effect of reduced weight bearing on lower-limb neuromuscular function compared to full weight bearing. Muscle activity, as measured via EMG, was significantly lower during reduced weight bearing when compared to upright standing for tibialis anterior, peroneus longus, vastus lateralis, and rectus femoris but not for medial and lateral head of gastrocnemius and vastus medialis. We found no difference in muscle activity between depth 1 and depth 5 on the ankle AMEDA for both positions. We found muscle tone and muscle stiffness measures, as determined via the MyontonPRO, were significantly lower in reduced weight bearing when compared to upright standing for all muscles assessed.

### 4.1. Ankle Somatosensation

Our hypothesis was supported in that reduced loading would decrease proprioceptive cues and lead to poorer accuracy in the AMEDA task. This suggests that ankle somatosensory acuity is reliant upon an upright, full weight-bearing posture. Research specifically on lower-limb somatosensory function in hypogravity is limited; however, it is clear that mobility and locomotion are negatively impacted in reduced gravity. Gait research studies have shown that in simulated 0.16 g, participants exhibit altered gait patterns, forward trunk tilt, and increased arm swing, which may indicate reduced sensory feedback from the feet and lower limbs [18,47]. In contrast, the upper limb seems less affected by reduced gravity, as previous proprioceptive research suggests finger somatosensory acuity and hand-position sense remain intact in simulated low gravity, compared to upright, full weight-bearing conditions [8,25].

The current study results of reduced lower-limb somatosensory function on the lunar wedge bed are comparable to our previous study, which compared ankle somatosensation in upright standing with horizontal supine lying (i.e., minimal plantar loading). In horizontal lying, the ankle somatosensory acuity was also reduced also by 0.02 AUC compared to the upright condition [25]. While it is clear that somatosensory acuity may be lower in reduced weight-bearing conditions, comparing both these study results implies that the degradation might not necessarily follow a linear pattern. Proprioception relies on mechanoreceptor cues and effort required to move a joint [8]. As muscle activity was reduced when on the lunar wedge bed, this may explain the poorer somatosensory acuity observed in this study. In low gravity, Roll et al., (1998) [48] suggested that the processing of these cues may be altered, with changes occurring at both the sensorimotor level and cognitive level. However, there is limited research on how somatosensory acuity might be affected by hypogravity, where there are likely small amounts of gravitational cueing occurring. Previous work investigating spatial orientation reported that one’s perception of upright occurs at ~0.15 g [49], while others reported it closer to 0.3 g [50]. This perception is closely tied to understanding body tilt and seems to be disrupted in astronauts moving on the Moon. Their inability to replicate Earth-like mobility on the Moon, even at 0.16 g, indicates that the sensory cues (vestibular and somatosensory) are insufficient for maintaining posture and balance. Our results raise the possibility that somatosensory acuity of the ankle may be impaired at 0.16 g and, possibly to a similar extent as observed in microgravity (µg) [25]. It could also reflect participants navigating the novel environment as they attempt to complete the AMEDA task in an unfamiliar position. Their patterns of movement may differ from what is considered normal under Earth’s 1 g conditions. This phenomenon has been described previously in gait and locomotion research on hypogravity. For example, when participants are asked to run on a treadmill under simulated lunar gravity (0.16 g), they often adopt a skipping style gait [18]. This gait pattern is thought to increase stability, suggesting that participants are attempting to stabilize themselves as they adjust their motor patterns in response to the new environment. Future studies would benefit from completing the ankle AMEDA task under a variety of loads in the same setting to map the trajectory of somatosensation at increasing loads of “g”.

The significant reduction in somatosensory acuity during the lunar wedge-bed position may have consequences for novel environments such as the Moon. While a reduction of 0.02 AUC score may appear minor, and the overlap of confidence intervals suggests ambiguity in the results, it could have functional implications for astronauts when multiple systems are compromised. The loss of 0.02 somatosensory acuity, combined with declines in the vestibular system, for example, could increase their risk of injury and falling. On Earth, diminished awareness of body position elevates the likelihood of injury or falls [51,52]. Moreover, ankle somatosensation plays a crucial role in maintaining balance, and impaired ankle somatosensory feedback is associated with conditions such as chronic ankle instability [53,54]. For astronauts, who are already adapting to the new and challenging environment, even a small reduction in ankle somatosensory acuity could substantially raise the risk of falls and injury [18]. A study by Orr et al., 2022 [55], assessed patterns of locomotion at Earth’s gravity (1 g), in simulated lunar gravity (0.16 g), and Mars’ gravity (0.38 g). They demonstrated asymmetric gait and inconsistent angles of dorsi flexion and plantar flexion in the lunar and Mars conditions compared to 1 g. The authors highlighted how these patterns could increase the risk of falls and injury in hypogravity environments. If both degrees of freedom—flexion/extension, as seen in Orr, et al., 2022 [55], and inversion/eversion, as seen in the current study—are negatively affected by reduced body loading, astronaut mobility issues and risk of recurring falls will ensue. The consequences of falls to astronauts whilst on the Moon may include injury, suit damage, suit contamination from excessive lunar dust, inability to retrieve lost items resulting from a fall, or inability to complete mission tasks [56]. It is not uncommon for an astronaut to adopt a jumping or hopping technique to mobilize on the Moon [29] but ankle kinematics can be an important predictor of knee kinematics. The way an ankle responds to a jump landing can influence a person’s risk of injuring their knee [57]. So, ankle somatosensory dysfunction could cascade into further gait and balance struggles. Further, in older adults, lower scores in the ankle AMEDA are associated with a higher risk of falls. Previous research has shown there to be a drop of 6% of somatosensory acuity between those with and without a history of falls [51]. Additionally, individuals with chronic ankle instability have an AUC 0.029 lower than those without instability, with repeated ankle AMEDA testing [58]. In contrast, higher AMEDA scores are correlated with athletic performance among adults [53]. For example, female football players and male basketball players, have AMEDA AUC scores of 0.71 and 0.70, respectively [59]. Furthermore, elite snow sport players and instructors have a mean AMEDA AUC score of 0.67 at the beginning of the snow season which improves by 0.04 AUC points by mid-season [60]. Targeted exercises to improve ankle function may be beneficial to reduce some of the risks to which astronauts are exposed.

Our results also raise questions about the potential implications of long-term exposure to hypogravity on somatosensory function. Previous work has shown that long-term exposure to microgravity (4–6 months) can have an impact on several physiological body systems, including a significant loss of bone density, muscle atrophy, and reduced cardiovascular fitness. When returning to Earth, these astronauts have poor locomotive control with changes in muscle activation, decreased spatial awareness, wide stance, and increased base of support [1,61]. Astronauts often face challenges adapting rapidly to changes in sensory cues when transitioning from zero gravity (0 g) back to Earth’s gravity (1 g), impacting precision, speed, and obstacle avoidance. This was shown by Mulavara et al., 2010 [9], during a purpose-built obstacle course assessment of astronauts’ locomotor function. Microgravity exposure of only 1–2 weeks can cause reduced coordination of the head and trunk, effectively changing the way astronauts walk [61]. However, it remains uncertain whether similar challenges arise with prolonged exposure to hypogravity to the somatosensation system or how this might impact an astronaut’s gait pattern and balance on the Moon (0.16 g). Given the acute exposure of this study, participants were likely learning how to navigate the reduced weight bearing. This might also explain the observed variability and overlap of confidence intervals in Figure 2 and Figure 3, where some individuals may have navigated the reduced weight bearing better than others. The experimental design was beneficial for capturing the impact of immediate hypogravity exposure to ankle somatosensation. However, within an hour or two, participants may have begun to adapt and develop new strategies to cope with the novel sensation, suggesting the results could have been different [18]. With even longer exposure and extended lower-limb de-loading, it is likely that they would experience physiological side effects consistent with those seen in chronic bed-rest studies and atrophy which would alter the results again. Further exploration of somatosensory changes in simulated hypogravity over several weeks or even months could provide valuable insights for space industry plans to establish a human presence on the Moon or Mars.

### 4.2. Neuromuscular Function: Muscle Activity

A reduction in muscle activity in the reduced weight-bearing condition compared to upright standing implies that the muscular demands to maintain standing in the simulated hypogravity position were much lower than in an upright, full weight-bearing position. The long-term effects of reduced muscle activity is muscle atrophy, which is associated with loss of bone mass, both of which are important indicators of astronaut health [55,62]. While previous micro- and hypogravity research indicated similar muscular degradation, the results of this study are unique in that the lower-limb neuromuscular changes were examined alongside the changes in ankle somatosensory acuity [10,33,63,64]. As a result, muscle spindles were likely to be active throughout the muscle activity recordings. As noted above, muscle spindles play a crucial role in proprioception and are thought to contribute to decreased somatosensation in space research, yet their activity is rarely assessed alongside a somatosensory task. Interestingly, in the current study, the gastrocnemius and vastus medialis were unaffected by the change in posture and reduced weight bearing. Further, these muscles displayed particularly widespread confidence intervals, highlighting the variability observed in the data. We hypothesize that some participants were attempting to stabilize their leg on the AMEDA platform as they resisted the vertical 1 g vector. This constant downward pull could potentially lead to undesirable side effects, which might explain the lack of change in muscle activity of gastrocnemius and vastus medialis between the body positions and high variability. However, it is important to note that participants, while only to a small degree, still had partial support under their feet, as they were still experiencing 16% of their body weight through their lower limbs. This suggests that some level of stability through the feet may assist in maintaining their leg position. If leg stabilization had been a major cause of concern, similar muscle patterns would likely have been observed in the other quadriceps muscles also.

In our previous study that assessed lower leg activity in a flat supine position, Gastrocnemius (along with other lower leg muscles) showed significantly less muscle activity when compared to upright standing [25]. Arguably, this flat supine position would require greater lower-limb muscle activation to maintain stability compared to the lunar wedge-bed setup; however, this was not the case. In the current study, gastrocnemius was subjected to partial loading on the lunar wedge bed (16% body weight) and there was no significant difference between positions, implying that 16% body weight was sufficient loading for gastrocnemius to continue activating at the same level as upright standing. However, tibialis anterior, peroneus longus, vastus lateralis, and rectus femoris were also subjected to the reduced loading and had a significant reduction in activity on the lunar wedge bed compared to upright standing. This suggests that gastrocnemius may have played a role in stabilizing the lower limb in some individuals when on the lunar wedge bed and therefore continued to activate, not to resist the vertical 1 g vector, but rather in a similar pattern to that in upright standing. As for vastus medialis, we observed considerable variability within this muscle when participants were upright and fully weight bearing, possibly explaining the lack of significant results with change in position. The primary function of vastus medialis is to stabilize the patella so the change in position (upright standing to supine lying on the lunar wedge bed) may not have significantly influenced vastus medialis’ role [65]. Additionally, the lack of significant difference in all muscle activity between inversion depths may be expected given the small difference between inversion depths 1 and 5 on the AMEDA (a 4-degree change). However, it is worth noting that despite this minimal difference, participants were sensitive to the change and somatosensory cues derived from it. This could be due to the EMG simply not being sensitive enough to identify differences in activity and being heavily influenced by the presence of noise. Alternatively, although beyond the scope of this paper, it highlights the importance of considering other sensory cues that impact somatosensation. Factors such as cutaneous feedback, athletic level, and likelihood of ankle instability are known to affect scores on the ankle AMEDA [53,58,66,67]. Exploring these factors within a hypogravity study could open avenues for interventions that could prove beneficial for astronauts.

### 4.3. Neuromuscular Function: Muscle Tone and Muscle Stiffness

The significant decrease in muscle tone and muscle stiffness when on the lunar wedge bed compared to upright standing may have implications for astronauts’ immediate exposure to hypogravity. An assessment of biomechanical properties can be a convenient and rapid indication of muscle health. This was demonstrated by Schoenrock et al., 2024 [22], who recorded muscle stiffness in twelve astronauts who were subjected to their routine workout regime when aboard the International Space Station (ISS). The researchers found that muscle stiffness was preserved in most muscles when on the ISS compared to when they were on Earth, except for tibialis anterior, which had significantly less stiffness despite the regular exercise. The decrease in tibialis anterior stiffness was comparable to the current study where the astronauts experienced a mean loss of 149 N/m, and our participants had a mean loss of 127 N/m on the lunar wedge bed compared to upright standing. In contrast to our study, Schoenrock et al. (2024) [22] found gastrocnemius stiffness to increase when on the ISS compared to on Earth. These patterns have also been observed in a dry immersion study where Amirova et al. (2021) [33] reported that a two-hour exposure to simulated microgravity via a dry immersion bed resulted in decreased tibialis anterior tone but increased gastrocnemius tone. While muscle tone and stiffness were reduced for all muscles in the current study when on the lunar wedge bed compared to upright standing, the magnitude of reduction for gastrocnemius was relatively small when compared to tibialis anterior. We found that the upper leg muscles had an even smaller reduction in muscle tone and stiffness when on the lunar wedge bed compared to upright standing. A major difference between these studies is that ours was conducted in simulated hypogravity, whereas Schoenrock et al. (2024) [22], and Amirova et al. (2021) [33] used real or simulated microgravity. Nevertheless, these studies demonstrate that various muscles respond differently to changes in loading whether it is simulated or real low gravity. Lower-limb muscles may therefore benefit from more targeted exercise regimes [22]. For example, the results of changes in muscle tone and muscle stiffness along with the changes in EMG muscle activity discussed above suggest muscle-targeted exercise regimes are perhaps less important for gastrocnemius and countermeasures need to consider this when training specific muscles.

### 4.4. Strengths and Limitations

This study has limitations: while we aimed to mimic the effects of hypogravity, we could not completely negate all effects of gravity (e.g., vestibular function, friction on the lunar wedge-bed timber board, and tactile sensation along the participants torso) within the constraints of the current study; we experienced high variability and excessive noise in EMG recordings which may have impacted the reliability of the data, and should be considered when interpreting the results, especially as it impacted the sample size of the EMG data [68]; an additional recording of muscle tone and stiffness during the AMEDA task instead of just at rest may have promoted further understanding of how these biomechanical properties reacted to the change in position.

The strengths of this study include the following: the experiment protocol was lodged with the open science framework to promote transparency and best research practice; the structure of the lunar wedge-bed system was considered successful due to its ease of use; participants were healthy and unrestricted so the large sample size is beneficial within space research but can be easily translated to the general adult population in areas such as health and exercise research.

## 5. Conclusions

We found ankle somatosensation and some lower-limb neuromuscular function was reduced in the supine lying on the lunar wedge-bed position when compared to upright standing. These results suggest that somatosensory and neuromuscular function are negatively affected by a rapid change in posture, even with a reduced degree of body loading. This has important implications for astronauts taking their first steps on the Moon, suggesting that a reduction in lower-limb function could increase the risk of injuries and falls as they adapt to the challenging environment. However, it is important to note that while all other muscles assessed were negatively impacted, the activity of gastrocnemius and vastus medialis remained unchanged by the change in posture. This could be attributed to the specific roles of these muscles, or excessive variability within the data. Additionally, there was no change in muscle activity across the various AMEDA inversion depths, possibly due to the limited sensitivity of the EMG. These findings suggest that further research is warranted to better understand neuromuscular patterns and explore other sensory cues which might impact somatosensation, particularly under reduced body loading and within real hypogravity conditions. Such knowledge can aid the development of countermeasures to negate the side effects of reduced body loading to ankle somatosensation and neuromuscular function. As space agencies prepare for upcoming lunar exploration, interventions to address and mitigate decreases in ankle somatosensory and lower-limb muscle function are crucial for ensuring astronaut safety and the success of missions.

## Figures and Tables

**Figure 1 brainsci-15-00443-f001:**
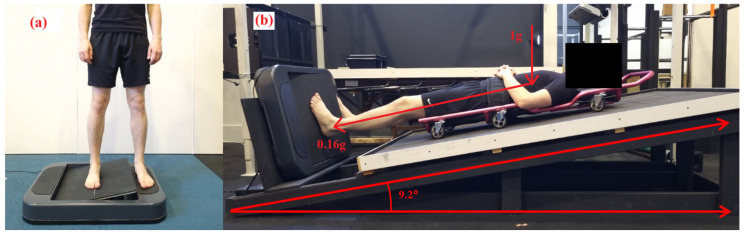
The AMEDA was used to assess participants’ ankle somatosensory acuity. The test required participants to place both feet on the AMEDA, either in (**a**) upright standing or (**b**) while lying on the lunar wedge bed. On the lunar wedge bed, (**b**) participants lay supine with their head elevated at approximately 9.2 degrees from horizontal. This position provided approximately 16% of the participant’s body weight through the plantar aspect of their feet (i.e., the vertical load, 1 g, is still present).

**Figure 2 brainsci-15-00443-f002:**
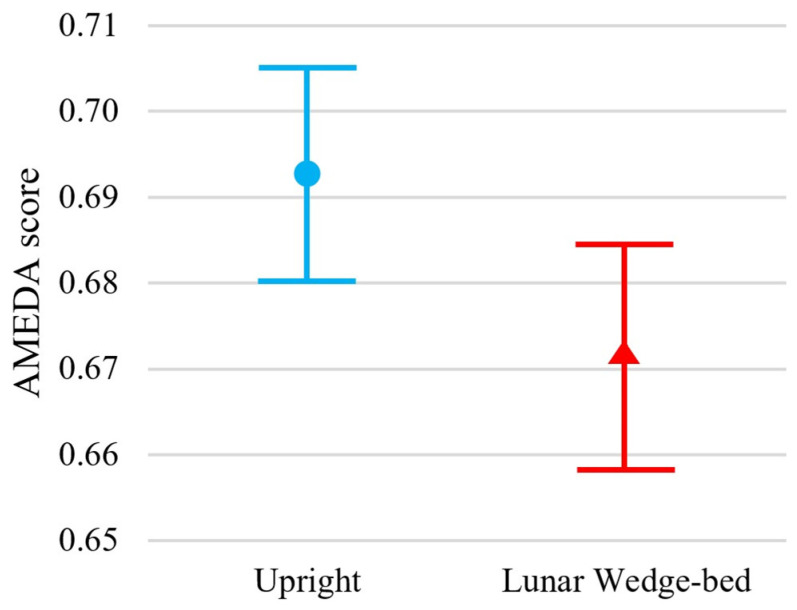
Results of the ankle AMEDA demonstrate greater performance in the upright standing position (represented by blue circle) compared to the head-elevated supine on lunar wedge-bed position (represented by red triangle). Mean scores are represented by an AUC score between 0.5 (score achieved by chance) and 1.0 (perfect score). Error bars represent 95% confidence interval.

**Figure 3 brainsci-15-00443-f003:**
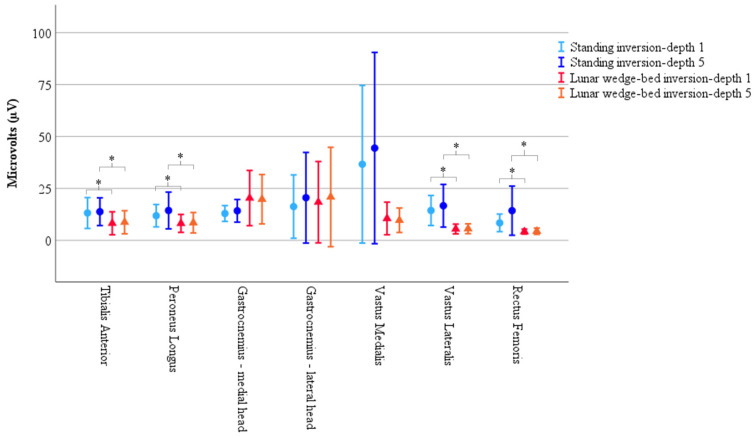
Results of muscle activity recorded via EMG demonstrate significantly less muscle activity in head-elevated supine on the lunar wedge bed (represented by triangles) compared to upright standing (represented by circles) for tibialis anterior, peroneus longus, vastus lateralis, and rectus femoris. Vastus medialis also had reduced activity on the lunar wedge bed compared to upright standing; however, it did not reach statistical significance. Activity of gastrocnemius increased when in upright standing compared to the lunar wedge bed but did not reach statistical significance. Results represented by the mean peak RMS values as measured in microvolts (µV). Error bars represent 95% confidence interval. * Denotes *p* < 0.05.

**Figure 4 brainsci-15-00443-f004:**
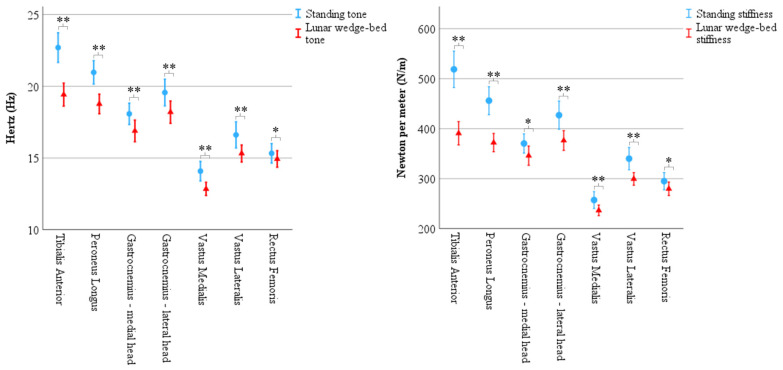
Results of biomechanical properties recorded via MyotonPRO demonstrate significantly less muscle tone and muscle stiffness in head-elevated supine on the lunar wedge bed (represented by triangles) compared to upright standing (represented by circles) for all muscles. Tone (**left**) is represented in Hertz (Hz) and stiffness (**right**) is represented in Newtons per meter (N/m). Error bars represent 95% confidence interval. * Denotes *p* < 0.05 and ** denotes *p* < 0.001.

**Table 1 brainsci-15-00443-t001:** EMG electrode sensor placement. Text derived from recommendations for sensor locations in upper and lower leg muscles (http://www.seniam.org/).

Muscle	EMG Electrode Placement Location
Tibialis anterior	1/3 on the line between the tip of the fibula and the tip of the medial malleolus
Peroneus longus	25% on the line between the tip of the head of the fibula to the tip of the lateral malleolus
Lateral head of gastrocnemius	1/3 of the line between the head of the fibula and the heel
Medial head of gastrocnemius	On the most prominent bulge of the muscle
Vastus medialis	80% on the line between the anterior spina iliaca superior and the joint space in front of the anterior border of the medial ligament
Rectus femoris	50% on the line from the anterior spina iliaca superior to the superior part of the patella
Vastus lateralis	2/3 on the line from the anterior spina iliaca superior to the lateral side of the patella

**Table 2 brainsci-15-00443-t002:** Results of muscle activity recorded via EMG for each muscle in upright standing and head-elevated supine on the lunar wedge bed positions for ankle AMEDA inversion depths 1 and 5. Results represented by peak RMS values as measured in microvolts (µV) displayed as mean ± standard deviation (M ± SD). *p* Values represent the follow up pairwise comparison between upright standing and head-elevated supine on the lunar wedge bed as the main effect of position was significant in the three-way ANOVA. * Denotes *p* < 0.05.

Muscle	AMEDA Inversion Depth	Standing EMG Activity (M ± SD)	Lunar Wedge Bed EMG Activity (M ± SD)	*p* Value
Tibialis anterior	Stop 1	13.2 ± 15.8	8.2 ± 11.8	0.004 *
	Stop 5	13.8 ± 14.3	8.7 ± 11.9	0.007 *
Peroneus longus	Stop 1	11.9 ± 11.5	8.2 ± 9.3	0.004 *
	Stop 5	14.4 ± 18.9	8.5 ± 10.5	0.009 *
Gastrocnemius, medial head	Stop 1	12.9 ± 8.1	20.3 ± 28.4	0.128
	Stop 5	14.2 ± 11.6	19.8 ± 25.4	0.191
Gastrocnemius, lateral head	Stop 1	16.3 ± 32.6	18.4 ± 41.8	0.429
	Stop 5	20.5 ± 46.6	20.9 ± 51.1	0.492
Vastus medialis	Stop 1	36.7 ± 81.1	10.6 ± 16.8	0.061
	Stop 5	44.4 ± 98.3	9.7 ± 12.5	0.057
Vastus lateralis	Stop 1	14.4 ± 15.4	5.5 ± 5.0	0.007 *
	Stop 5	16.7 ± 22.0	5.6 ± 5.1	0.019 *
Rectus femoris	Stop 1	8.4 ± 9.0	4.2 ± 2.5	0.027 *
	Stop 5	14.3 ± 25.3	4.4 ± 3.2	0.043 *

**Table 3 brainsci-15-00443-t003:** Results of muscle tone recorded via MyotonPRO for each muscle in upright standing and head-elevated supine on the lunar wedge bed positions. Mean ± standard deviation (M ± SD) with simple pairwise comparison *p* value shown for each. Tone is represented in Hertz (Hz) and was reduced for all muscles when on the lunar wedge bed compared to upright standing. * Denotes *p* < 0.05 and ** denotes *p* < 0.001.

Muscle	Standing Tone (M ± SD)	Lunar Wedge Bed Tone (M ± SD)	*p* Value
Tibialis anterior	22.7 ± 3.7	19.4 ± 2.9	<0.001 **
Peroneus longus	21.0 ± 2.9	18.8 ± 2.5	<0.001 **
Gastrocnemius, medial head	18.1 ± 2.7	16.9 ± 2.7	<0.001 **
Gastrocnemius, lateral head	19.6 ± 3.4	18.2 ± 2.8	<0.001 **
Vastus medialis	14.1 ± 2.5	12.8 ± 1.7	<0.001 **
Vastus lateralis	16.6 ± 3.3	15.3 ± 2.1	<0.001 **
Rectus femoris	15.3 ± 2.5	14.9 ± 2.1	0.049 *

**Table 4 brainsci-15-00443-t004:** Results of muscle stiffness recorded via MyotonPRO for each muscle in upright standing and head-elevated supine on the lunar wedge-bed positions. Mean ± standard deviation (M ± SD) with simple pairwise comparison *p* value shown for each. Stiffness is represented in Newtons per meter (N/m) and was reduced for all muscles when on the lunar wedge bed compared to upright standing. * Denotes *p* < 0.05 and ** denotes *p* < 0.001.

Muscle	Standing Stiffness (M ± SD)	Lunar Wedge Bed Stiffness (M ± SD)	*p* Value
Tibialis anterior	518.7 ± 132.1	391.0 ± 84.6	<0.001 **
Peroneus longus	456.1 ± 101.3	372.1 ± 66.3	<0.001 **
Gastrocnemius, medial head	370.4 ± 69.6	346.0 ± 69.6	0.002 *
Gastrocnemius, lateral head	427.0 ± 102.5	376.3 ± 71.4	<0.001 **
Vastus medialis	257.3 ± 61.7	236.5 ± 38.7	<0.001 **
Vastus lateralis	340.1 ± 81.0	299.5 ± 46.0	<0.001 **
Rectus femoris	295.0 ± 62.8	280.0 ± 49.3	0.002 *

## Data Availability

The study protocol was uploaded to the Open Science Framework prior to data collection (Marchant, 2023 [36]). The raw data supporting the conclusions of this article will be made available by the authors, without undue reservation.

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
