# Peer review of "Ankle Somatosensation and Lower-Limb Neuromuscular Function on a Lunar Gravity Analogue"

_brainsci, 2025, doi:10.3390/brainsci15050443_

Round 1

Reviewer 1 Report

Comments and Suggestions for Authors

General comments

This manuscript examines the effects of reduced weight-bearing, simulating lunar gravity, on ankle somatosensation and neuromuscular function. It is a timely and relevant study, particularly with space exploration on the horizon. The research is novel and uses an innovative lunar wedge-bed setup, but the manuscript would benefit from a clearer explanation of why this method was chosen over other common techniques in gravity-related research. The methods are appropriate, but the sample size of 55 participants is relatively small, limiting the statistical power of the findings. The rationale for statistical choices, particularly in handling outliers, should be more clearly explained.

While the writing is generally clear, some sections are difficult to follow due to technical jargon, and certain terms should be defined for clarity. The discussion section could delve more into the broader implications of the findings, particularly how these results could inform rehabilitation strategies or future space missions. Additionally, the connection between somatosensation and muscle activity is not fully explored—clarifying this relationship would improve the understanding of the observed results.

Overall, the study provides valuable insights but requires revisions for clarity, a more detailed statistical explanation, and a stronger discussion of the implications. With these improvements, the manuscript could make a significant contribution to the field.

Specific Comments

  1. Abstract
    • Line 9-11: The phrase "adverse effects of low gravity to human physiology" should be reworded to "adverse effects of low gravity on human physiology" for grammatical accuracy.
    • Line 16-17: The phrase "to simulate hypogravity" might be clearer as "to simulate conditions of hypogravity." Consider rephrasing for precision.
  2. Introduction
    • Lines 43-46: The discussion about the challenges astronauts face in low gravity environments is important, but the statement "errors can have catastrophic outcomes" could be better supported with specific examples or citations.
    • Lines 51-56: The introduction to hypogravity is relevant, but the reference to microgravity and its implications is slightly confusing. You may want to make the distinction clearer between hypogravity and microgravity conditions in the context of the study.
  3. Methods
    • Lines 173-175: The description of the lunar wedge-bed design and its angle should include more detail on how the angle was determined. The calculation referenced here (FN = FWSinθ) could be expanded with a brief explanation to clarify its relevance.
    • Lines 187-191: The explanation of the lunar wedge-bed setup is adequate, but could benefit from visual clarification. Since the study design relies on the bed for the simulation of lunar gravity, the specifics of how the weight is reduced could be explained in greater detail for transparency.
    • Lines 239-245: The choice of EMG electrodes and muscle groups assessed is well-reasoned. However, it would be beneficial to mention the specific placement or anatomical landmarks used for electrode positioning, especially for the Gastrocnemius muscle.
  4. Results
    • Lines 359-361: The presentation of the AMEDA results could be clearer. Consider breaking down the statistical findings into more digestible parts, highlighting the significant differences and their implications in the narrative.
    • Lines 374-376: The EMG results show some variability. It might be helpful to explain how the variability was managed in the analysis and whether it affects the reliability of the findings.
    • Line 380: The statement that "muscle activity was significantly reduced on the lunar wedge-bed compared to upright standing" is clear, but it would be better to provide a brief explanation of why some muscles showed no significant difference (e.g., Gastrocnemius and Vastus Medialis).
  5. Discussion
    • Lines 469-471: The comparison to previous studies is useful, but further elaboration on how the results align or differ from the existing literature would strengthen the discussion.
    • Lines 474-480: The statement "The loss of 0.02 somatosensory acuity" should be expanded to explain why this slight reduction may still be significant. While the discussion of astronaut gait patterns on the Moon is relevant, the connection between somatosensory acuity and potential falls/injuries could be made more explicit.
  6. Conclusion

Lines 649-651: The conclusion is concise, but it could be stronger if the potential applications of the findings are highlighted more explicitly. The implications for countermeasures and interventions should be emphasized, particularly in the context of lunar missions.

Reviewer 2 Report

Comments and Suggestions for Authors

Dear Authors,
Thank you for the scientific work provided. The article demonstrates interesting and, in principle, expected results on the effect of simulated microgravity on the somatosensory system. The current work performed by the Authors is a consistent continuation of previously published results, which at this stage, the original results submitted to the journal Brain Sciences deserve consideration.

  1. Since we are talking about the potential application of the obtained results in the preparation of astronauts before sending them into weightlessness, it may make sense to consider several subsamples with different mathematically expected body masses of the subjects. The results may differ dramatically.
  2. As shown in Figure 2, the confidence intervals for Upright and Lunar Wedge-bed overlap, which makes it impossible to provide an unambiguous assessment of their difference from the point of view of mathematical statistics for comparative analysis.
  3. A similar remark for the next figure (Figure 3). The error is many times greater than the result obtained. At the same time, for some positions the error limits are negative. It is necessary to revise.
  4. There are minor typos throughout the text, such as misplaced indexes (line 184) , etc.

Round 2

Reviewer 1 Report

Comments and Suggestions for Authors

The authors have successfully addressed my comments.

Reviewer 2 Report

Comments and Suggestions for Authors

I am quite satisfied with the provided explanations and changes in the 
scientific content of the manuscript. I recommend the work for further processing.